# STC-ViT: Spatio Temporal Continuous Vision Transformer for Medium-range Global Weather Forecasting

## Abstract

Operational Numerical Weather Prediction (NWP) systems rely on computationally expensive physics-based models. Recently, transformer models have shown remarkable potential in weather forecasting achieving state-of-the-art results. However, traditional transformers discretize spatio-temporal dimensions, limiting their ability to model continuous dynamical weather processes. Moreover, their reliance on increased depth to capture complex dependencies results in higher computational cost and parameter redundancy. We address these issues with **STC-ViT**, a Spatio-Temporal Continuous Vision Transformer for weather forecasting. STC-ViT integrates a Fourier Neural Operator (FNO) for global spatial operators with a transformer-parameterized Neural ODE for continuous-time dynamics, yielding a space–time continuous model of weather forecasting. Our proposed method achieves competitive forecasting performance even with a shallow, single-layer transformer encoder, and scales further with depth as shown in our analysis (Section 5.5). STC-ViT generates complete forecast trajectories with an inference speed of only 0.125 seconds and achieves strong medium-range forecasting skill on 1.5° WeatherBench 2 as compared to state-of-the-art data-driven and NWP models trained on higher-resolution data, with lower data and compute costs. We also provide detailed empirical analysis on model's performance with respect to denser time grids, higher-accuracy ODE solvers, and deeper transformer stacks. Our code is available at `https://anonymous.4open.science/r/STCViT-CC8B`.

## 1 Introduction

Weather forecasting can be formulated as a continuous spatio-temporal forecasting problem in which atmospheric state variables such as temperature, pressure, humidity, and wind evolve together across both space and time typically expressed as time-dependent partial differential equations (PDEs) (Couairon et al., 2024). Operational weather forecasting systems solve these PDEs relying on physics based Numerical Weather Prediction (NWP) models which require high computational resources (Palmer et al., 2005; Andersson, 2022). Recent works have shown that using deep learning models as surrogates to model complex multi-scale spatio-temporal phenomena can lead to training-efficient models (Gupta & Brandstetter, 2022). Vision Transformer by Dosovitskiy et al. (2020) has emerged as one such model (Lessig et al., 2023; Nguyen et al., 2023b).

However, transformers in weather forecasting often exhibit structural limitations. Traditional ViT architectures process input data by dividing it into distinct patches and discrete time intervals, posing a significant challenge in learning the continuous evolution inherent in atmospheric dynamics (Fonseca et al., 2023). Because standard encoder-decoder transformers operate in discrete time, they are typically trained separately for each lead time or rely on multi-head outputs that still require the network to implicitly learn distinct temporal dynamics for each horizon, which leads to parameter duplication.

Moreover, transformers generally require very deep architectures with large attention stacks to capture multi-scale spatiotemporal processes, since attention mechanisms do not naturally encode the smooth, continuous evolution governed by physical dynamics. This depth amplifies training cost, memory footprint, and sensitivity to data sparsity. As a result, traditional transformers struggle to model the continuous, long-horizon,

behavior characteristic of weather fields without substantial architectural engineering or horizon-specific training regimes.

In this work, we address the issue of **non-continuity** and **the reliance on high-depth encoder stacks** in Vision Transformer (ViT) models. We build upon the core ideas of temporal continuity introduced by Neural Ordinary Differential Equation (NODE) (Chen et al., 2018), spatial continuity by Fourier Neural Operator (FNO) (Li et al., 2020) and ViT (Dosovitskiy et al., 2020). We propose STC-ViT which leverages the continuous learning paradigm to effectively learn the complex spatio-temporal changes even from weather data recorded at coarser resolution. This approach alleviates the need for excessively deep transformer stacks by modeling dynamics in continuous time, reducing architectural depth without sacrificing spatiotemporal expressiveness. The idea is to parameterize transformer depth as a continuous-time flow by modeling the block as a time-conditioned vector field and integrating tokens with an ODE solver. The continuous-time dynamics are integrated for each sample and merged with patch-level spectral context by projecting band-limited FNO features and spherical-harmonic encoding onto the patch grid. This fusion equips each ViT token with globally informed spectral structure and smooth continuous-time evolution, producing coherent spatiotemporal representations across the embedding space. This results in stronger long-range coherence alongside temporal continuity and coordinate awareness.

We show that with these architectural modifications, shallow ViTs are as powerful as some of the deeper and temporally continuous weather forecasting models. STC-ViT with depth 1 trained on 5.625° resolution data outperforms deterministic ViT with depth 8 and Neural ODE based architectures. Additionally, STC-ViT trained on 1.5° performs competitively with state-of-the-art models while operating under lower data volumes, compute overhead, training and inference times as shown in Section 6. Finally, we present a comprehensive capacity–performance analysis, showing that forecast skill improves with higher-accuracy ODE solvers, and deeper transformer blocks. In summary, our contributions are as follows:

1. We propose a novel way to model weather as a continuous spatio-temporal flow, treating the Transformer encoder as a time-conditioned vector field and integrating tokens with an ODE solver to realize continuous depth with explicit accuracy/latency control.

2. We introduce a spectral branch that pairs FNO with a spherical-harmonic encoder to produce band-limited, topology-aware global features improving cross-patch coherence.

3. We perform extensive experiments on both WeatherBench and WeatherBench 2 to show the competitive performance of STC-ViT against state-of-the-art forecasting models.

## 2 Related Work

### 2.1 Numerical Weather Prediction

Integrated Forecasting System (IFS) (ECMWF, 2023) is the operational NWP based weather forecasting system coupling a spectral-transform dynamical core with a suite of physics parametrizations to solve the full three-dimensional, time-dependent equations of fluid motion, thermodynamics, moisture and radiation on a rotating sphere. IFS runs at horizontal resolutions which generates forecasts at a high resolution of 9km. IFS combines data assimilation and an Earth-system model to generate accurate forecasts using high-performance supercomputers. In contrast, data-driven methodologies have recently achieved competitive accuracy against NWP models at substantially lower inference cost.

### 2.2 Data-driven Weather Forecasting

WeatherBench by Rasp et al. (2020) provides a benchmark platform to evaluate data-driven systems for effective development of weather forecasting models. Current state-of-the-art data-driven forecasting models are mostly based on Graph Neural Networks (GNNs) and Transformers. Keisler (2022) implemented a message passing GNN based forecasting model which was further extended by GraphCast by Lam et al. (2023) which used multi-mesh GNN to achieve state-of-the-art results. FourCastNet (FCN) by Kurth et al. (2023) used Fourier Neural Operator (FNO) with a ViT backbone and was reported to be 80,000 times faster

than NWP models. FourCastNetV3 by Bonev et al. (2025), a probabilistic version of FCN is shown to be more stable and accurate than its deterministic counterpart.

Several more transformer based models emerged including Pangu-Weather (Bi et al., 2023). Pangu-Weather uses a large-scale vision transformer (ViT)-based structure capable of capturing both spatial patterns and temporal evolution through attention mechanisms. It outperformed several traditional NWP models on medium-range weather prediction benchmarks indicating that transformer architectures can achieve high accuracy without explicit physics constraints, purely driven by data. ClimaX by Nguyen et al. (2023a) presented as a foundational model excels in its ability to integrate various sources of climatological data seamlessly, including heterogeneous observational inputs and multi-modal datasets. Additionally, FengWu (Chen et al., 2023a), FuXi (Chen et al., 2023b), Stormer (Nguyen et al., 2023b) and ArchesWeather (Couairon et al., 2024) all showcased remarkable capabilities by effectively modeling long-range spatiotemporal dependencies and leveraging innovative attention mechanisms, positional embeddings, hierarchical architectures, and adaptive resolutions, consistently performing strongly in both short- and medium-range forecasting.

While being highly accurate and showcasing remarkable scaling capabilities, these models are discrete space-time models and do not account for the continuous dynamics of weather system. Verma et al. (2023) proposed ClimODE which used Neural ODE to incorporate a continuous-time process that models weather evolution and advection, enabling it to capture the dynamics of weather transport across the globe effectively. However it currently yields less precise forecast results compared to state-of-the-art models, offering significant potential for further enhancements. Further, Kochkov et al. (2023) proposed Neural GCM, which integrates a differentiable solver with neural networks resulting in physically consistent models.

## 3 Background

### 3.1 Neural ODE

Neural ODEs are the continuous time models which learn the evolution of a system over time using Ordinary Differential Equations (ODE) (Chen et al., 2018). The key idea behind Neural ODE is to model the derivative of the hidden state using a neural network. Consider a hidden state $h(t)$ at time $t$. In a traditional neural network, the transformation from one layer to the next could be considered as moving from time $t$ to $t + 1$. In Neural ODEs, instead of discrete steps, the change in $h(t)$ over time is defined by an ordinary differential equation parameterized by a neural network:

$$\frac{dh(t)}{dt} = f(h(t), t, \theta) \tag{1}$$

where $\frac{dh(t)}{dt}$ is the derivative of the hidden state with respect to time, $f$ is a neural network with parameters $\theta$ and $t$ is the time variable, allowing the dynamics of $h(t)$ to change with time. Neural ODEs are considered as a continuous depth version of ResNets. Consider a ResNet with layers updating the hidden state as:

$$h_{t+1} = h_t + f(h_t, \theta_t) \tag{2}$$

In the limit, as the number of layers goes to infinity and the updates become infinitesimally small, this equation resembles the Euler method for numerical ODE solving, where:

$$h(t + \Delta t) = h(t) + \Delta t \cdot f(h(t), t, \theta) \tag{3}$$

Reducing $\Delta t$ to an infinitesimally small value transforms the discrete updates into a continuous model described by the ODE given earlier. Computing the output of a Neural ODE requires numerically integrating $f$ forward in time from $t_0$ to a target time $t_1$. Gradients with respect to parameters are obtained either by reverse-mode differentiation through the solver's operations or, more memory-efficiently, via the adjoint sensitivity method (Chen et al., 2018). Either way, the integration itself is performed with numerical ODE

solvers such as Euler, Runge-Kutta, or more sophisticated adaptive methods, which can efficiently handle the potentially complex dynamics encoded by $f$.

## 3.2 Spectral Branches

**Fourier Neural Operator (FNO):** FNOs learn mappings between function spaces by operating in the Fourier domain (Li et al., 2020). For an input field $u: \mathbb{T}^d \to \mathbb{R}^C$ one layer computes

$$v(x) = W_0 u(x) + \mathcal{F}^{-1}\big(R(k)\,\widehat{u}(k)\,\mathbf{1}_{\|k\| \le K}\big)(x). \tag{4}$$

$\widehat{u}(k) = \mathcal{F}[u], R(k)$ are learned mode-wise matrices, $K$ is a bandlimit and $W_0$ is a pointwise mixer. This realizes a learned Fourier multiplier resulting in global mixing with $O(N \log N)$ Fast Fourier Transform (FFT) cost, producing a band-limited, spatially continuous field. It leads to efficient capture of large-scale modes and a natural prior for smooth geophysical fields.

**Spherical Harmonics (SH):** On the sphere $S^2$, fields expand in spherical harmonics $Y_{\ell m}$ (Dai & Xu, 2013)

$$
\begin{aligned}
a_{\ell m} &= \int_{S^2} u(\Omega)\,\overline{Y_{\ell m}(\Omega)}\,d\Omega, \\
v(\Omega) &= W_0\,u(\Omega) + \sum_{\ell=0}^{L}\sum_{m=-\ell}^{\ell} Y_{\ell m}(\Omega)\,R_\ell\,a_{\ell m}.
\end{aligned}
\tag{5}
$$

A learned, degree-wise spectral filter $R_\ell$ with bandlimit $L$ yields a topology-aware, continuous global representation with no dateline seam or polar distortion. SH provides a coordinate-invariant positional basis and clean control of spectral content. In weather forecasting, SH/SFNO matches spherical geometry, stabilizes long rollouts, and enables physics-aligned regularization directly in coefficient space.

# 4 Continuous-Depth Spectral–Token Encoder for Global Weather Forecasting

## 4.1 Problem Statement

We formulate weather forecasting as a spatio-temporal continuous problem, modeling atmospheric fields as functions on the sphere that evolve in continuous time rather than as discrete step-to-step maps. Given an initial multivariate atmospheric state (initial condition) $x_0 \in \mathbb{R}^{B \times C_{\text{in}} \times H \times W}$ on the sphere where $B$ is the batch size, $C_{\text{in}}$ is the number of input (weather) variables, $H \times W$ is the latitude-longitude grid, we learn continuous-time latent evolution that advances a tokenized representation of the atmospheric state via a Transformer-parameterized ODE while retaining the spherical topology through harmonic positional encoding and injecting global spectral context via a truncated Fourier layer.

## 4.2 Spherical Harmonic as Positional Encoder

We use spherical harmonics as a topology-aware positional encoding at the level of patch centers. Let the input grid have height $H$ and width $W$, and let the patch size be $p$. The number of non-overlapping patches is $N_p = (H/p)\,(W/p)$. We construct patch-center latitudes $\{\theta_i\}_{i=1}^{H/p} \subset [-\frac{\pi}{2}, \frac{\pi}{2}]$ and longitudes $\{\phi_j\}_{j=1}^{W/p} \subset [0, 2\pi)$ over the padded grid and select the centers with stride $p$, their tensor-product yields the $N_p$ spherical coordinates $\{(\theta_n, \phi_n)\}_{n=1}^{N_p}$. Given a bandlimit $L \in \mathbb{N}$, we consider all degrees and orders $(\ell, m)$ with $0 \le \ell \le L$ and $-\ell \le m \le \ell$. For each patch center with latitude-longitude $(\theta_n, \phi_n)$, we evaluate spherical harmonics using colatitude $\vartheta_n = \frac{\pi}{2} - \theta_n \in (0, \pi)$ and azimuth $\phi_n \in [0, 2\pi)$:

$$Y_{\ell m}(\vartheta, \phi) = N_{\ell m}\,P_\ell^{|m|}(\cos\vartheta)\,e^{im\phi},$$

where $P_\ell^{|m|}$ are the associated Legendre functions and $N_{\ell m}$ is the standard normalization. For each patch $n$ and pair $(\ell, m)$, we form real-valued features by taking real and imaginary parts,

$$S_{n,(\ell,m,\text{Re})} = \Re\{Y_{\ell m}(\vartheta_n, \phi_n)\}, \qquad S_{n,(\ell,m,\text{Im})} = \Im\{Y_{\ell m}(\vartheta_n, \phi_n)\},$$

and stack these across all $(\ell, m)$ to obtain the positional matrix

$$S \in \mathbb{R}^{N_p \times 2M}, \qquad M = \sum_{\ell=0}^{L} (2\ell + 1).$$

A linear projection maps $S$ to the embedding dimension $D$, followed by normalization, yielding the spherical-harmonic positional tokens given in equation 6. These tokens depend on patch locations on the sphere and result in a smooth, topology-aware positional basis that anchors the data-dependent features.

$$T_{\text{SH}} = \text{LN}(S\,W_{\text{SH}}) \in \mathbb{R}^{N_p \times D}, \qquad W_{\text{SH}} \in \mathbb{R}^{(2M) \times D}. \tag{6}$$

### 4.3 Fourier Spectral Layer

We apply a single, band-limited Fourier layer that mixes the input globally in the frequency domain.

**Frequency projection.** Given the input field

$$x_0 \in \mathbb{R}^{B \times C_{\text{in}} \times H \times W},$$

we take a 2-D Fourier transform along the last two (spatial) axes to obtain

$$\widehat{x}_0 = \mathcal{F}[x_0] \in \mathbb{C}^{B \times C_{\text{in}} \times H \times (W/2+1)}.$$

Here $B$ is batch size, $C_{\text{in}}$ the number of input variables, and $(H, W)$ the latitude-longitude grid. The hat denotes complex Fourier coefficients. Because $x_0$ is real-valued, only the non-redundant half-spectrum is kept along longitude.

**Low-frequency truncation and learned mixing.** We keep a spectral window of low/mid frequencies of size $(M_h, M_w)$ and set all other modes to zero. This hard band-limit suppresses high-frequency noise and stabilizes training. Inside this block, each frequency $(i, j)$ is transformed *mode-wise*. The input channels are linearly mixed into $D$ output channels using learned real/imaginary multipliers $W^{\text{re}}, W^{\text{im}} \in \mathbb{R}^{D \times C_{\text{in}} \times M_h \times M_w}$. Concretely, for batch index $b$, output channel $o$, and mode $(i, j)$, we form the complex output coefficient

$$\widehat{y}_{b,o,i,j} = \sum_{c=1}^{C_{\text{in}}} \left( \widehat{x}^{\text{re}}_{b,c,i,j} + i\,\widehat{x}^{\text{im}}_{b,c,i,j} \right) \left( W^{\text{re}}_{o,c,i,j} + i\,W^{\text{im}}_{o,c,i,j} \right),$$

where $\widehat{x}^{\text{re}}, \widehat{x}^{\text{im}}$ are the real/imag parts of the input spectrum. (Equivalently, the real and imaginary parts follow the standard complex multiply, the imaginary equation is omitted here for brevity.) This yields the band-limited, learned spectrum $\widehat{y}$ used in the inverse transform. Indices: $b$ (batch), $c$ (input channel), $o$ (output channel $1 \leq o \leq D$), and $(i, j)$ (frequency mode with $0 \leq i < M_h$, $0 \leq j < M_w$). Inverse Fourier transform is applied to obtain a band-limited spatial feature map as shown in equation 7.

$$\phi_{\text{FNO}} = \mathcal{F}^{-1}[\widehat{y}] \in \mathbb{R}^{B \times D \times H \times W}. \tag{7}$$

We average $\phi_{\text{FNO}}$ over each non-overlapping $p \times p$ patch $\Omega_n$ to align with patch tokens.

$$T_{\text{FNO}}[b, n, :] = \frac{1}{p^2} \sum_{(u,v) \in \Omega_n} \phi_{\text{FNO}}[b, :, u, v] \in \mathbb{R}^D, \quad n = 1, \ldots, N_p, \tag{8}$$

stacking to $T_{\text{FNO}} \in \mathbb{R}^{B \times N_p \times D}$, where $N_p = (H/p)\,(W/p)$ is the number of patches and $D$ is the embedding dimension. The resulting $T_{\text{FNO}}$ given in equation 8 provides each token with a summary of long-range, low-frequency content, which complements location-aware spherical positional codes and improves the stability and consistency of subsequent continuous-time token dynamics.

### 4.4 Continuous Transformer Encoder

**Fusion and initial condition.** The tokens produced by each branch are summed to form the latent initial condition for our transformer-parameterised NODE, given by equation 9.

$$z(0) \;=\; T_{\mathrm{disc}} \;+\; T_{\mathrm{pos}} \;+\; T_{\mathrm{FNO}} \;+\; T_{\mathrm{SH}} \;\in\; \mathbb{R}^{B \times N_p \times D} \tag{9}$$

where $T_{\mathrm{disc}}$ are data-dependent patch tokens from the ViT stem with per-variable patch embedding and cross-variable attention aggregation following ClimaX (Nguyen et al., 2023a), $T_{\mathrm{pos}} \in \mathbb{R}^{N_p \times D}$ is a learnable positional embedding initialised with 2-D sinusoidal values (Dosovitskiy et al., 2020) that provides a generic euclidean positional basis over the patch grid. $T_{\mathrm{FNO}}$ are Fourier-pooled tokens from the spectral branch, and $T_{\mathrm{SH}}$ are spherical-harmonic positional tokens that supply a complementary topology-aware basis on the sphere. $T_{\mathrm{pos}}$ and $T_{\mathrm{SH}}$ play complementary roles: the former encodes flat patch-grid coordinates as in standard ViTs, while the latter respects the curvature and pole/dateline geometry of the latitude-longitude domain. Here $B$ is batch size, $N_p$ the number of patches, and $D$ the embedding dimension.

**Sinusoidal temporal embeddings.** We encode time $t$ as a $D$-dimensional sinusoidal vector given in equation 10.
$$e(t) = \big( \sin(\omega_0 t), \cos(\omega_0 t), \dots, \sin(\omega_{K-1} t), \cos(\omega_{K-1} t) \big) \in \mathbb{R}^D \tag{10}$$

which provides a smooth, multi-scale embedding across temporal scales. This conditioning is shared across all patches at the same $t$ and varies smoothly as $t$ changes, enabling interpolation across lead times.

**Time-conditioned vector field.** We treat the fused token sequence $z(0)$ given in equation 9 as the *initial state* of a Neural ODE. The time variable $t$ enters the dynamics through its sinusoidal embedding $e(t) \in \mathbb{R}^D$, which is broadcast uniformly across all $N_p$ patches and injected inside $f_\theta$ as described below. The continuous-time evolution is

$$\frac{dz(t)}{dt} \;=\; f_\theta\big(z(t), e(t)\big), \qquad z(0) = T_{\mathrm{disc}} + T_{\mathrm{pos}} + T_{\mathrm{FNO}} + T_{\mathrm{SH}}. \tag{11}$$

Rather than treating attention as a discrete sequence operator, we reinterpret it as a learnable vector field evolving over a latent spatio-temporal manifold. We instantiate $f_\theta$ as a time-aware transformer dynamical operator built from $L$ pre-norm ViT blocks (Dosovitskiy et al., 2020). Each block performs multi-head self-attention for global context propagation through query-key-value projections, a position-wise feed-forward transformation that captures non-linear local feature refinements, and residual and normalisation pathways that stabilise integration across time. Before every block, the time embedding $e(t)$ is added to the current latent state, so the temporal coordinate conditions both the attention and feed-forward computations at every depth.

Concretely, setting $h_0 = z(t)$, the dynamics function $f_\theta$ at time $t$ is computed by equation 12: for $\ell = 1, \dots, L$ we form a time-conditioned input $\tilde{h}_{\ell-1} = h_{\ell-1} + e(t)$, then apply pre-norm self-attention and an MLP with residual connections, and finally normalise the output of the last block.

$$
\begin{aligned}
\tilde{h}_{\ell-1} &= h_{\ell-1} + e(t), \\
u_\ell &= \tilde{h}_{\ell-1} + \mathrm{Attn}\big(\mathrm{LN}(\tilde{h}_{\ell-1})\big), \\
h_\ell &= u_\ell + \mathrm{MLP}\big(\mathrm{LN}(u_\ell)\big), \qquad \ell = 1, \dots, L, \\
f_\theta\big(z(t), e(t)\big) &= \mathrm{LN}(h_L) \;\in\; \mathbb{R}^{B \times N_p \times D}.
\end{aligned}
\tag{12}
$$

For the depth-1 configuration used in our main results, the loop collapses to a single iteration. The scaling-analysis configurations in Section 5.5 use $L = 8$, in which case $e(t)$ is added before each of the eight blocks. The trajectory $z(t)$ is then obtained by integrating this ODE over the requested lead times $t_{1:T}$ as shown in equation 13. This results in a latent trajectory $z(t_\ell) \in \mathbb{R}^{B \times N_p \times D}$ at each lead, which is subsequently decoded to the latitude-longitude grid.

$$\{z(t_\ell)\}_{\ell=1}^T \;=\; \text{odeint}\Big(f_\theta,\; z(0),\; \tau \cdot \{0, t_1, \ldots, t_T\}\Big), \qquad \tau = 10^{-2}. \tag{13}$$

Here $\tau$ is a fixed scalar that rescales the physical lead times (expressed in hours) into the dimensionless time variable used by the solver. This rescaling keeps the magnitudes of $t$ and $\|f_\theta\|$ on a comparable order, which improves the conditioning of the ODE and the stability of the Euler/RK4 updates without altering the model's expressive capacity.

**Decoding to the physical grid.** For each lead $t_\ell$, we reshape tokens to a patch grid and interpolate back to the full spatial resolution before a per-pixel head:

$$
\begin{aligned}
z(t_\ell) &\xrightarrow{\text{reshape}} z_g(t_\ell) \in \mathbb{R}^{B \times D \times H_p \times W_p} \\
&\xrightarrow{\text{upsample}} \tilde{z}(t_\ell) \in \mathbb{R}^{B \times D \times H \times W} \\
&\xrightarrow{\text{head}} \hat{x}_{t_\ell} \in \mathbb{R}^{B \times C_{\text{out}} \times H \times W},
\end{aligned}
\tag{14}
$$

where $H_p = H/p$ and $W_p = W/p$ denote the patch-grid dimensions, and $H \times W$ is the original latitude-longitude resolution.

We reconstruct forecasts on the physical grid by first interpolating the latent patch grid to the target latitude–longitude resolution using a smooth bilinear mapping defined by the given coordinate arrays. The resulting per-location latent vectors of dimension (D) are then mapped to the required prognostic variables by a lightweight MLP predictor.

The additive fusion produces an initial latent $z(0)$ that already combines (i) data-dependent patch content, (ii) global, band-limited spectral context, and (iii) topology-aware spherical location. The ODE then provides a *continuous-time* evolution of these tokens, controlled by the solver and time grid, while the decoder cleanly maps each latent state back onto the physical latitude–longitude grid at the requested lead times. The complete architectural pipeline of STC-ViT is given in Figure 1.

## 5 Experiments and Results

### 5.1 Dataset

We train STC-ViT on ERA5 dataset Hersbach et al. (2018; 2020) provided by the European Center for Medium-Range Weather Forecasting (ECMWF). We compare STC-ViT against several weather forecasting models by training it at two different resolutions of $5.625°$ (32 x 64 grid points) provided by WeatherBench Rasp et al. (2020) and $1.5°$ (121 x 240 grid points) provided by WeatherBench2 (Rasp et al., 2024). All training details and hyperparameters are given in Section A.

### 5.2 Evaluation Metrics

We used Root Mean Square Error (RMSE) equation 15 and Anomaly Correlation Coefficient (ACC) equation 16 metrics to evaluate our model's predictions.

$$RMSE = \frac{1}{N} \sum_{k=1}^{N} \sqrt{\frac{1}{H \times W} \sum_{i=1}^{H} \sum_{j=1}^{W} L(i)(\hat{X}_{k,i,j} - X_{k,i,j})^2} \tag{15}$$

where $H \times W$ is the spatial resolution of the weather input and $N$ is the number of total samples used for training or testing. $L(i)$ is the latitude-weight that accounts for the smaller area of grid cells near the poles $L(i) = \frac{\cos \phi_i}{\frac{1}{H} \sum_{i'=1}^{H} \cos \phi_{i'}}$, where $\phi_i$ is the latitude (in radians) of grid row $i$.

$$ACC = \frac{\sum_{k,i,j} L(i)\, \hat{X}'_{k,i,j} X'_{k,i,j}}{\sqrt{\sum_{k,i,j} L(i)\hat{X}'^{2}_{k,i,j} \sum_{k,i,j} L(i)X'^{2}_{k,i,j}}} \tag{16}$$

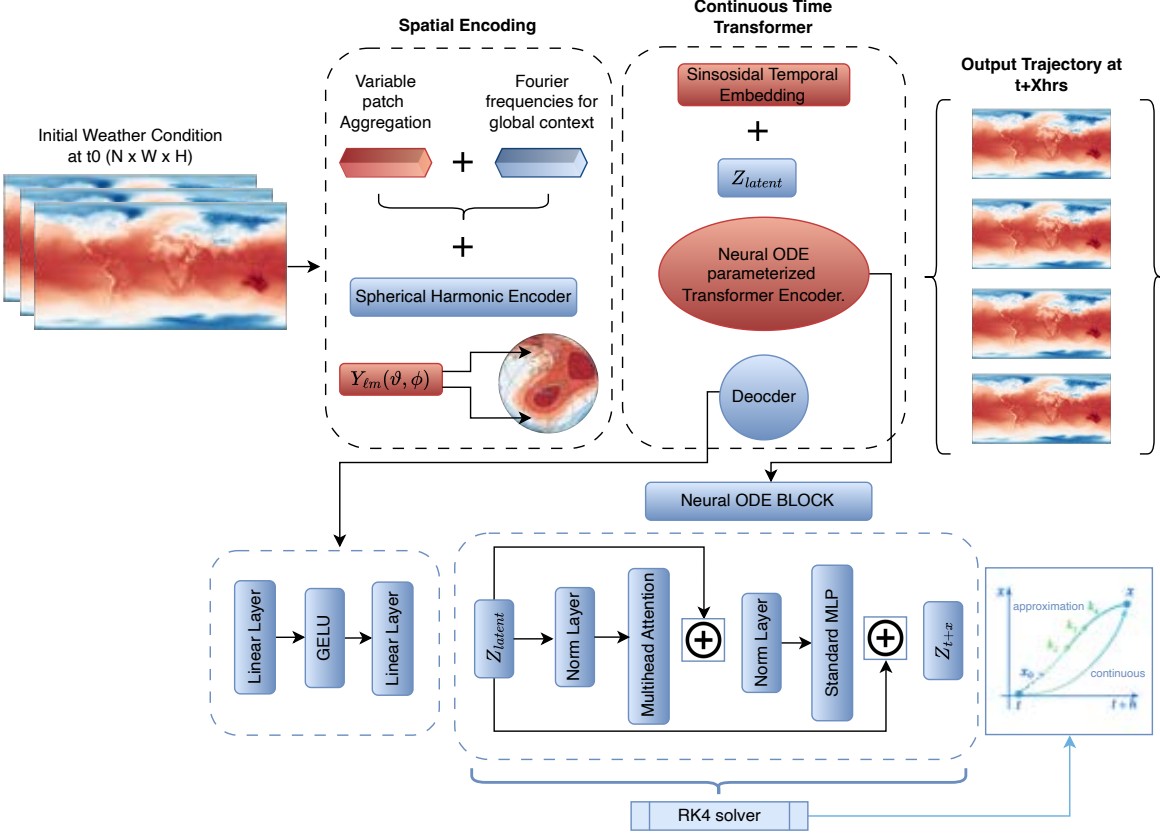

Figure 1: Complete architectural pipeline of STC-ViT

Where $\hat{X}' = \hat{X} - C$ and $X' = X - C$ and $C$ is the climatology, computed as the temporal mean over the training set, $C = \frac{1}{N} \sum_k X_k$.

### 5.3 WeatherBench

We train STC-ViT on hourly data from 1979-2015 for training, 2016 for validation and 2017-2018 for testing phase. We compare STC-ViT with ClimaX, and ClimODE on ERA5 dataset at 5.625° resolution provided by WeatherBench (Rasp et al., 2020). To ensure fairness, we retrained ClimaX with depth 8 from scratch without any pre-training. ClimODE and STC-ViT are both trained on 'euler' solver and STC-ViT with depth 1.

STC-ViT outperforms non-pretrained ClimaX (with depth 8) and continuous-time ClimODE at all lead times which shows that adding global context in local patch driven transformer architecture along with continuous depth parameterization of encoder derives improved feature extraction by mapping the changes occurring between successive time steps leading to improved prediction scores. The RMSE and ACC results are shown in Table 1. The visual results are shown in Section B.

### 5.4 WeatherBench2

To keep training consistent with WeatherBench 2, we utilize the training data from 1979 to 2018, validation data from 2019, and test data from 2020. We train STC-ViT on hourly data for following variables: T2m, u10 and v10 wind components and mean sea-level pressure (MSLP) along with five atmospheric variables: geopotential height (Z), temperature (T), U and V wind components, and specific humidity (Q). These

Table 1: RMSE and ACC results of STC-ViT at 5.6° compared against ClimODE (ODE based) and ClimaX (non-pretrained) trained on ERA5 at 5.625° resolution

| Variable | Lead Time (hrs.) | RMSE (Lower is better) | | | ACC (Higher is better) | | |
|---|---|---|---|---|---|---|---|
| | | STC-ViT | ClimODE | ClimaX | STC-ViT | ClimODE | ClimaX |
| z500 $(m^2/s^2)$ | 12 | **129.89** | 142.44 | 203.9 | **0.99** | 0.98 | 0.98 |
| | 18 | **151.07** | 183.73 | 236.36 | **0.99** | 0.98 | 0.97 |
| | 24 | **166.85** | 222.99 | 263.83 | **0.99** | 0.97 | 0.96 |
| | 36 | **221.96** | 309.68 | 343.15 | **0.97** | 0.95 | 0.94 |
| | 72 | **403.53** | 518.01 | 590.77 | **0.92** | 0.88 | 0.80 |
| | 144 | **714.24** | 813.6 | 828.9 | **0.71** | 0.61 | 0.60 |
| u10 $(m/s)$ | 12 | **1.43** | 1.75 | 1.55 | **0.94** | 0.90 | 0.92 |
| | 18 | **1.54** | 1.93 | 1.82 | **0.93** | 0.88 | 0.90 |
| | 24 | **1.66** | 2.12 | 2.04 | **0.91** | 0.85 | 0.87 |
| | 36 | **1.96** | 2.49 | 2.51 | **0.88** | 0.79 | 0.79 |
| | 72 | **2.83** | 3.29 | 3.45 | **0.73** | 0.66 | 0.55 |
| | 144 | **3.98** | 4.12 | 4.14 | **0.43** | 0.35 | 0.36 |
| v10 $(m/s)$ | 12 | **1.47** | 1.81 | 1.57 | **0.93** | 0.89 | 0.92 |
| | 18 | **1.57** | 1.97 | 1.83 | **0.92** | 0.88 | 0.90 |
| | 24 | **1.69** | 2.16 | 2.48 | **0.91** | 0.85 | 0.87 |
| | 36 | **1.99** | 2.54 | 2.52 | **0.88** | 0.78 | 0.79 |
| | 72 | **2.88** | 3.35 | 3.53 | **0.73** | 0.63 | 0.52 |
| | 144 | **4.10** | 4.34 | 4.45 | **0.38** | 0.32 | 0.29 |
| T2m (K) | 12 | **1.44** | 2.93 | 1.76 | **0.96** | 0.82 | 0.94 |
| | 18 | **1.46** | 2.15 | 1.89 | **0.96** | 0.91 | 0.93 |
| | 24 | **1.45** | 2.05 | 1.82 | **0.96** | 0.92 | 0.93 |
| | 36 | **1.61** | 2.71 | 2.36 | **0.95** | 0.83 | 0.89 |
| | 72 | **1.87** | 2.85 | 2.63 | **0.93** | 0.80 | 0.86 |
| | 144 | **2.52** | 3.37 | 3.28 | **0.87** | 0.78 | 0.83 |
| T850 (K) | 12 | **1.28** | 1.38 | 1.52 | **0.96** | 0.95 | 0.95 |
| | 18 | **1.34** | 1.52 | 1.68 | **0.96** | 0.95 | 0.94 |
| | 24 | **1.40** | 1.70 | 1.79 | **0.96** | 0.93 | 0.93 |
| | 36 | **1.56** | 2.07 | 2.08 | **0.95** | 0.90 | 0.90 |
| | 72 | **2.10** | 2.72 | 2.85 | **0.90** | 0.85 | 0.81 |
| | 144 | **3.16** | 3.72 | 3.95 | **0.77** | 0.77 | 0.70 |

atmospheric variables are considered at 13 pressure levels: 50, 100, 150, 200, 250, 300, 400, 500, 600, 700, 850, 925, 1000 hPa. We also compare our results with IFS-HRES (Andersson, 2022), which is state-of-the-art forecasting model at high resolution of 0.1°. We also compare STC-ViT against GraphCast and Pangu-Weather which are trained at a higher resolution of 0.25°.

Following the WB2 protocol (Rasp et al., 2024), all models are evaluated on a common 1.5° grid. Forecasts and ground truth from 0.25° (GraphCast, Pangu-Weather) and 0.1° (IFS-HRES) baselines are re-gridded to 1.5° before scoring.

STC-ViT shows competitive performance and outperforms IFS-HRES, Pangu-Weather and GraphCast at lead times greater than 7 days particularly for temperature variables. Note that this performance comes with STC-ViT trained with transformer depth 1, achieving an inference speed of 0.125 seconds. Our scaling analysis in Section 5.5 shows that increasing transformer depth to 8 sees a substantial increase in performance of STC-ViT.

We observe that although STC-ViT demonstrates consistent forecasting skill at medium-range lead times, it underperforms at shorter lead times. This discrepancy is due to the coarse temporal resolution of its trajectory generation scheme. Specifically, training on sparse lead-time grids hinder the model's ability to capture fine-scale temporal dependencies and nonlinear interactions that evolve continuously over time. We show in our scaling analysis study that when the model is trained on denser temporal grids, it exhibits improved generalization and smoother temporal consistency, indicating that the continuous-time formulation effectively captures short-range temporal transitions as well.

We believe that STC-ViT could also benefit from high resolution training and richer embedding space and denser time grids which we will explore in future work. The ACC and RMSE results are shown in Figure 2.

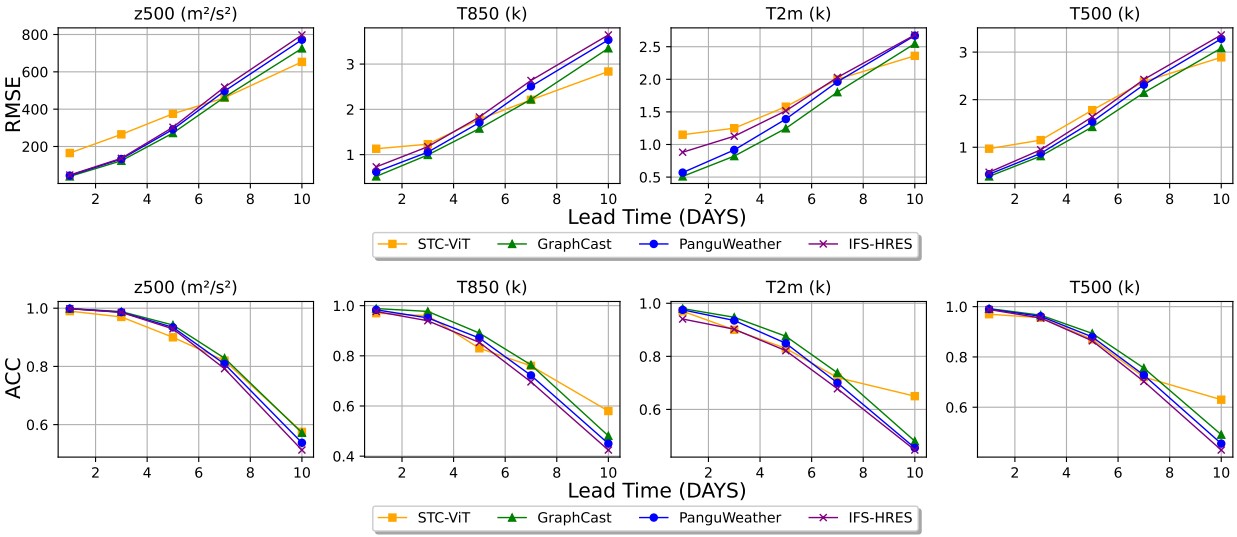

Figure 2: RMSE and ACC comparison of STC-ViT trained at 1.5° with GraphCast, Pangu-Weather at 0.25° and IFS-HRES at 0.1° resolution data for lead times ranging from 1 to 10 days

## 5.5 Scaling Analysis

We conducted extensive scaling experiments to evaluate the sensitivity of our model's forecasting skill to numerical integration schemes, model depth, and temporal grids. First, we compared the performance of the Euler integrator and the Runge-Kutta 4 (RK4) solver for temporal integration within the Neural ODE block. The higher-order RK4 method improved the model's forecasting accuracy by approximately 5%, indicating that more precise numerical integration enhances the stability of temporal dynamics and gradient propagation. Next, we analyzed the effect of transformer depth by comparing configurations with depth-1 and depth-8 encoder-decoder layers. Increasing depth significantly improved representational capacity and long-range dependency modeling, yielding roughly a 20% increase in overall forecasting performance. Finally, we compared the sparse vs. denser time grid performance. The model trained on denser time grids performs considerably better than the one trained on a sparse time grid, leading to an accuracy vs. training-time trade-off. The results of scaling analysis are shown in Figure 3.

## 5.6 Ablation Studies

We performed three ablation studies to highlight the importance of each component of STC-ViT. All studies were performed on WeatherBench data of resolution 5.625°. The RMSE results of ablation studies are shown in Figure 4

**Vanilla Vision Transformer:** We trained a basic ViT architecture with depth 1 on ERA5 dataset. Compared with STC-ViT, ViT under performs for prediction accuracy showing the superiority of continuous models in weather forecasting systems.

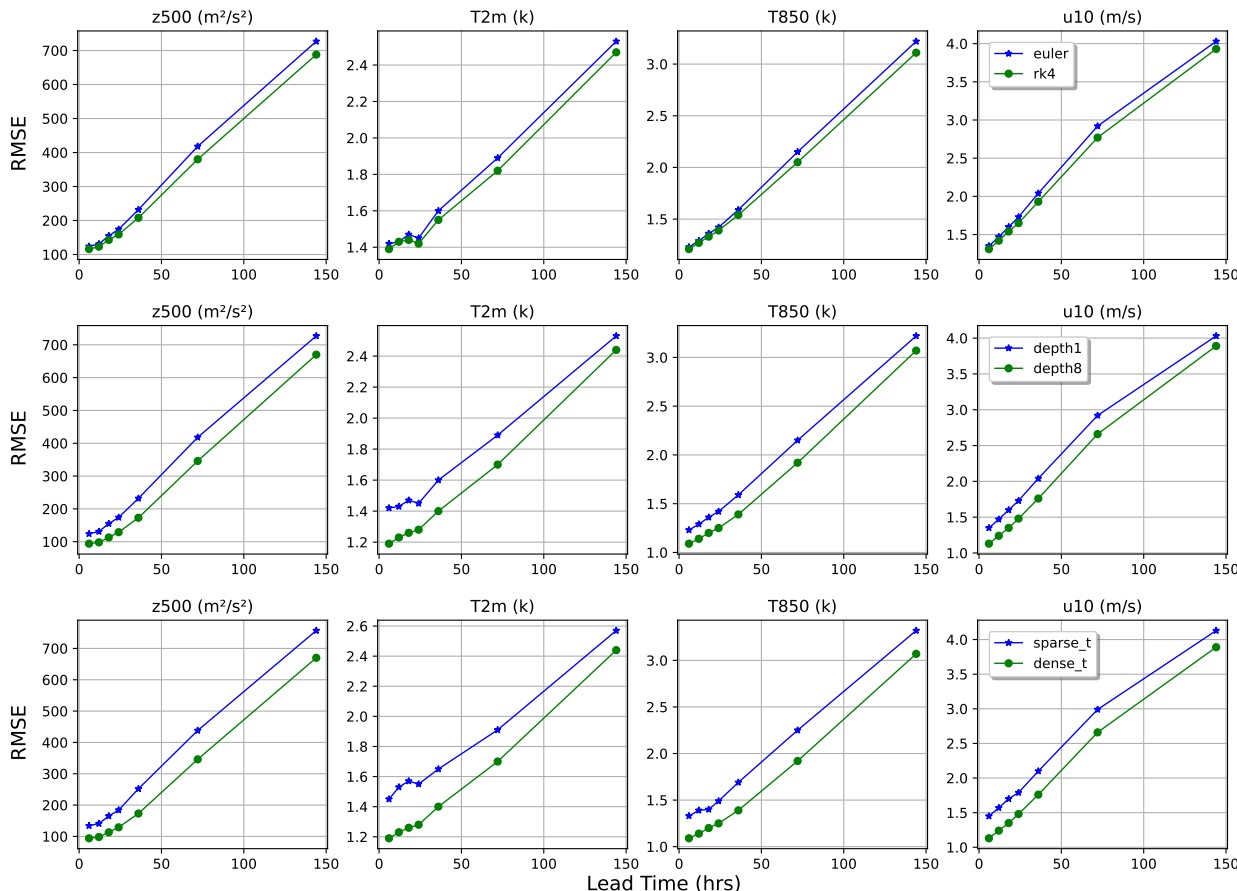

Figure 3: Scaling analysis shows STC-ViT consistently improves as we employ denser time-grids, higher accuracy order adaptive solvers and increase transformer depth.

**FNO Head:** For this study, we trained the model by adding FNO head to learn the global context in addition to the local patch context. The performance improved as compared to Vanilla ViT, highlighting the importance of FNO which learns global spatial interactions by operating in the Fourier domain, where each mode captures information across the entire field.

**Neural ODE parameterized by a Transformer Encoder:** Finally, we replaced the discrete transformer encoder with a continuous parameterized neural ode which improves the performance by a margin emphasizing the efficiency of continuous-time representations for modeling long-range dependencies and smooth spatiotemporal dynamics.

# 6 Conclusion and Future Work

In this paper, we present STC-ViT, a novel technique designed to capture continuous dynamics of weather data. STC-ViT achieves competitive results in weather forecasting which shows that vision transformers can model the continuous nature of spatio-temporal dynamic systems with careful parameterizations.

While STC-ViT performs competitively with state-of-the-art data-driven weather forecasting models, it is important to address its limitations in weather forecasting systems. STC-ViT is inherently based on transformer architecture which has a limitation of having higher training times when scaled to higher resolutions. Further, the deterministic nature of our approach does not account for uncertainties which can produce unrealistic results. Extending STC-ViT to a probabilistic model can be addressed in future work.

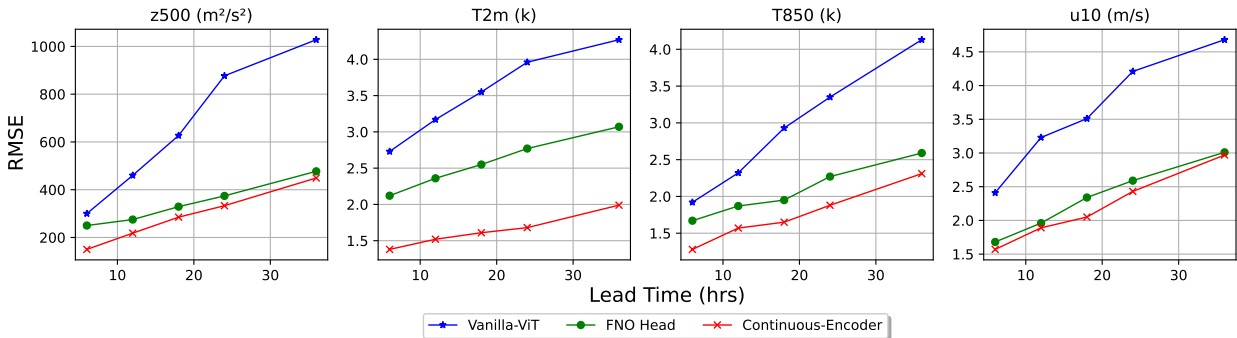

Figure 4: Ablation studies showing how adding each component to the network improves the performance of STC-ViT. The vanilla ViT shows the worst performance, highlighting the major drawback of transformer architectures

It is also important here to address the biases resulting from training on one dataset. The ERA5 dataset, while a robust and widely used weather dataset also has inherent limitations. Potential biases in data-sparse regions, challenges in representing local phenomena, and inconsistencies in observational continuity may impact model generalisability. To address this, future work will explore: (i) data augmentation to synthetically increase dataset diversity, (ii) integration with diverse datasets, such as regional or event-specific datasets, to mitigate geographical and climatic biases, and (iii) scaling STC-ViT to better accommodate multi-modal high-resolution training and evaluation. Finally, addressing the black-box problem of STC-ViT can shed light on model learning insights which is equally important for climate science community.

**Broader Impact Statement**

Our research contributes to addressing climate change by focusing on modeling the continuous dynamics of weather and climate through a continuous shallow Transformer-based architecture. Unlike conventional deep, discrete transformer models requiring per-lead-time training, the proposed continuous-time formulation captures long-range spatio-temporal dependencies within a unified, low-parameter ODE framework. By integrating ML to improve accuracy while optimizing computational efficiency, we can create a sustainable and inclusive approach to weather forecasting that serves the global community more effectively. The training and inference times are shown in Table 2 Additionally, training STC-ViT at higher resolution on more diverse datasets and focusing on high resolution regional forecasts as a future work can directly contribute to enhanced climate resilience, enabling societies to better anticipate and adapt to extreme weather events such as hurricanes, droughts, and floods. Further, extending the study to cater for longer lead times of few months and seasonal can help in disaster preparedness, as timely and precise forecasts allow governments and organisations to implement early warning systems, plan evacuations, and allocate resources effectively, thereby mitigating loss of life and property.

Table 2: Run-Time comparison against several data-driven weather forecasting models

| Model | Parameters | Training Time | Inference Time | Train Device |
|---|---|---|---|---|
| Pangu-Weather | 256M | 64 days | ≈ 14 seconds | 192 NVIDIA Tesla-V100 GPUs |
| GraphCast | 37M | 4 weeks | <60 seconds | 32 TPUs |
| ClimaX (non-pretrained) | 107M | ≈ 2 days | ≈ 1 min | 8 V100 |
| STC-ViT (5.625/1.5°) | 26.2M/156M | 2 days / 15 days | 31ms / 125ms | 4 V100 / 2 H200 |

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

# A    Training Details

We consider weather forecasting as a continuous spatio-temporal forecasting problem, i.e., an initial weather condition at $t_0$ of shape $N \times H \times W$ at time $t$ is fed to the model which outputs a continuous trajectory of $T' \times N' \times H' \times W'$ where $T'$ is the future time step trajectory, $N'$ is the number of predicted weather variables and $H' \times W'$ is the lat-lon grid. The complete training hyperparameters of the model are given in Table 3.

## A.1    Software and Hardware Requirements

We use PyTorch (Paszke et al., 2019), Pytorch Lightning (Falcon, 2019), torchdiffeq (Chen et al., 2018), and xarray (Hoyer & Hamman, 2017) to implement our model. We use 2 NVIDIA H200 devices for training STC-ViT at 1.5° and 4 NVIDIA Tesla Volta V100-SXM2-32GB for the training at resolution of 5.625°.

Table 3: Hyperparameters of STC-ViT

| Hyperparameters | Meaning | Value |
|---|---|---|
| p | patch size | 2 |
| heads | number of attention heads | 16 |
| depth | number of transformer layers | 1 |
| decoder_depth | number of MLP layers in the per-pixel head | 2 |
| solver | ODE solver | euler (main results); rk4 (scaling analysis) |
| dimension | hidden dimensions | 1024 |
| dropout | dropout rate | 0.1 |
| lr | learning rate | 5e-5 |
| batch_size | batch size | 8 for 5.625°, 2 for 1.5° |
| $\beta 1$, $\beta 2$ | decay rate of AdamW optimizer | 0.9, 0.999 |
| optim | optimizer | AdamW |
| fno_modes | number of fourier coefficients | $(8, 8)$ for 5.625°, $(30, 30)$ for 1.5° |
| $\ell_{max}$ | number of harmonic frequency bands | 8 for 5.625°, 30 for 1.5° |
| $\tau$ | ODE time-rescaling factor | $10^{-2}$ |
| norm | data normalization | z_score |
| epochs | total training epochs | 150 for 5.625°, 50 for 1.5° |

# B    Qualitative results

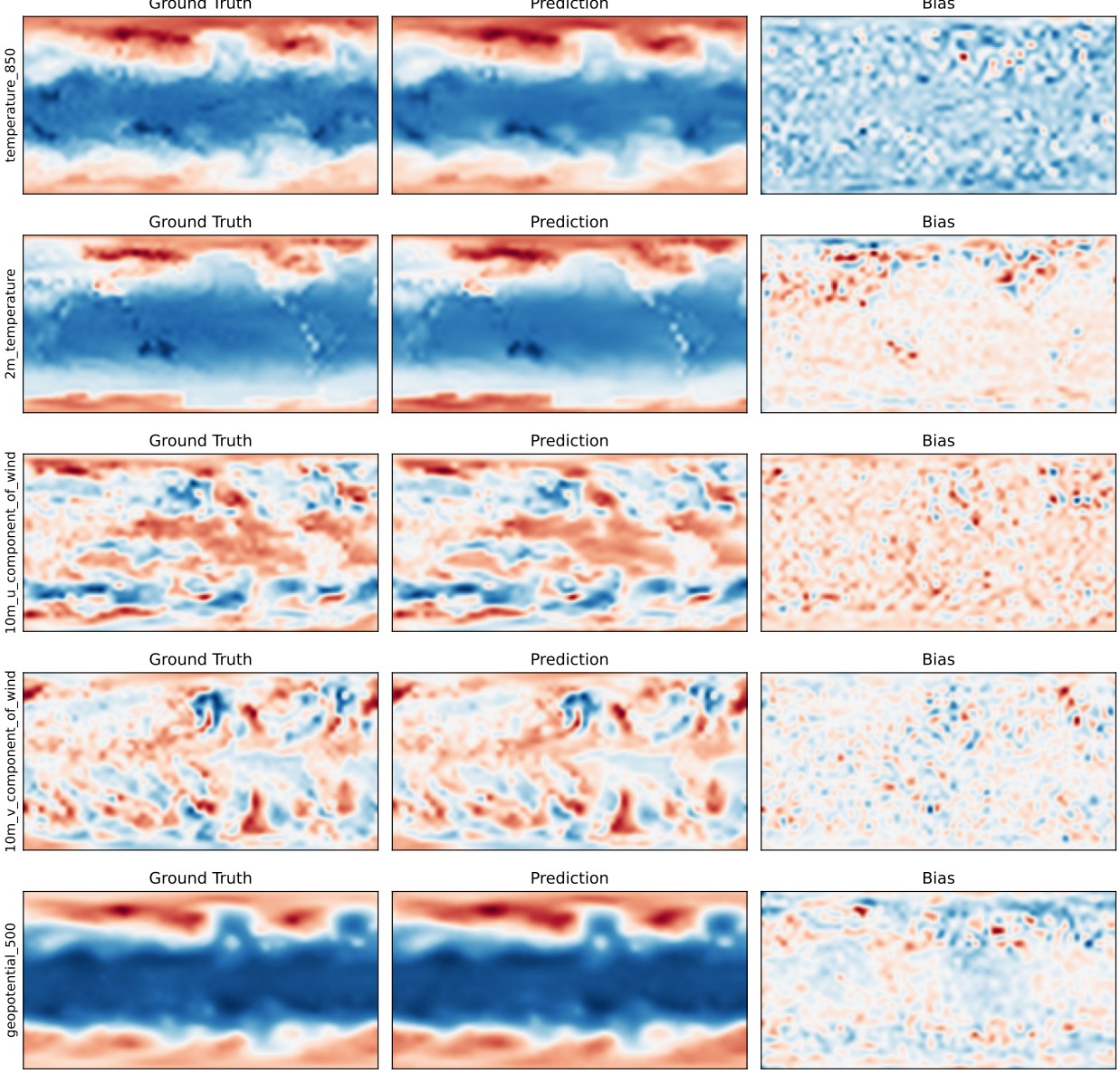

Figure 5: 6hr forecast results of STC-ViT

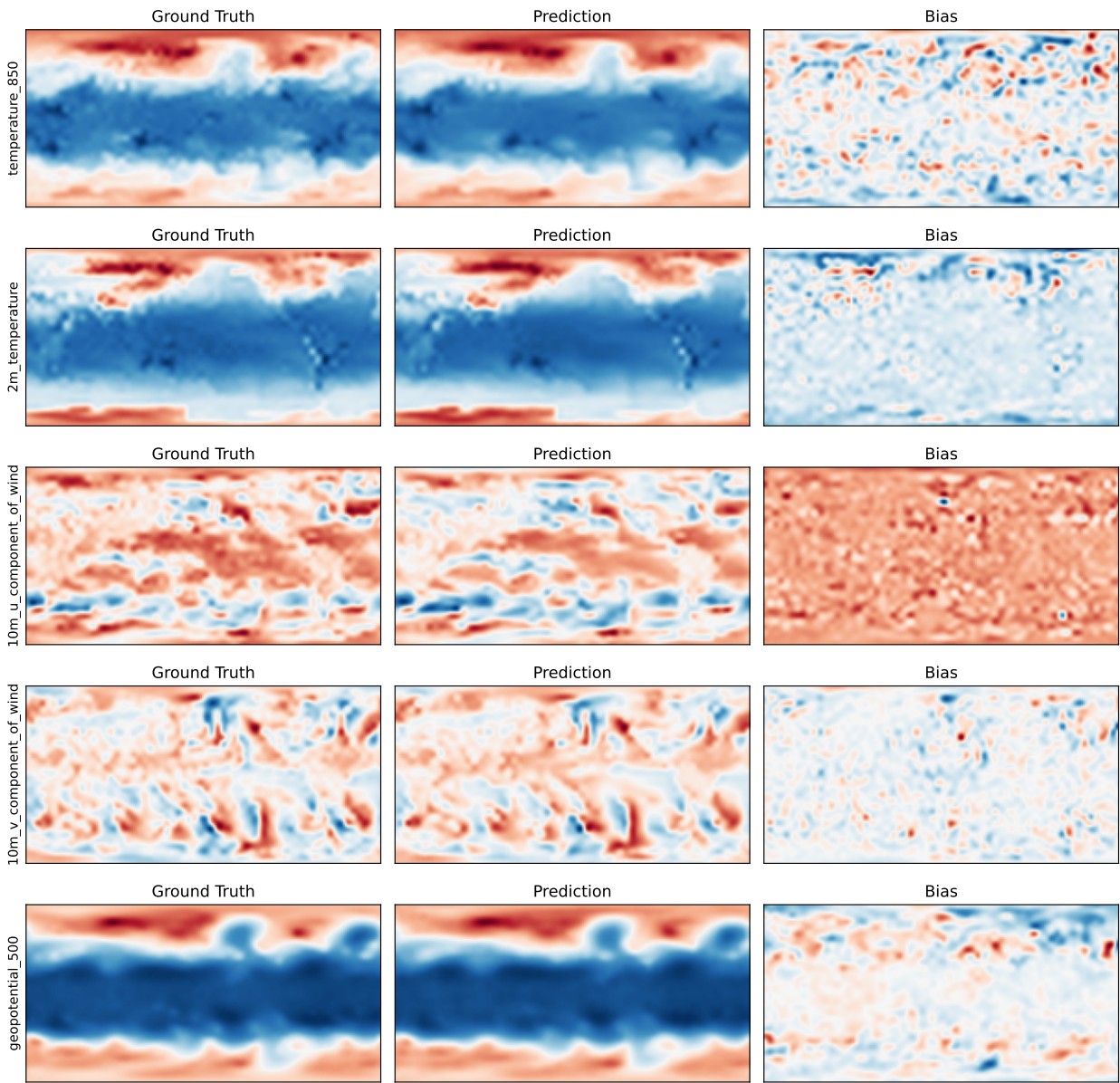

Figure 6: 1 day forecast results of STC-ViT

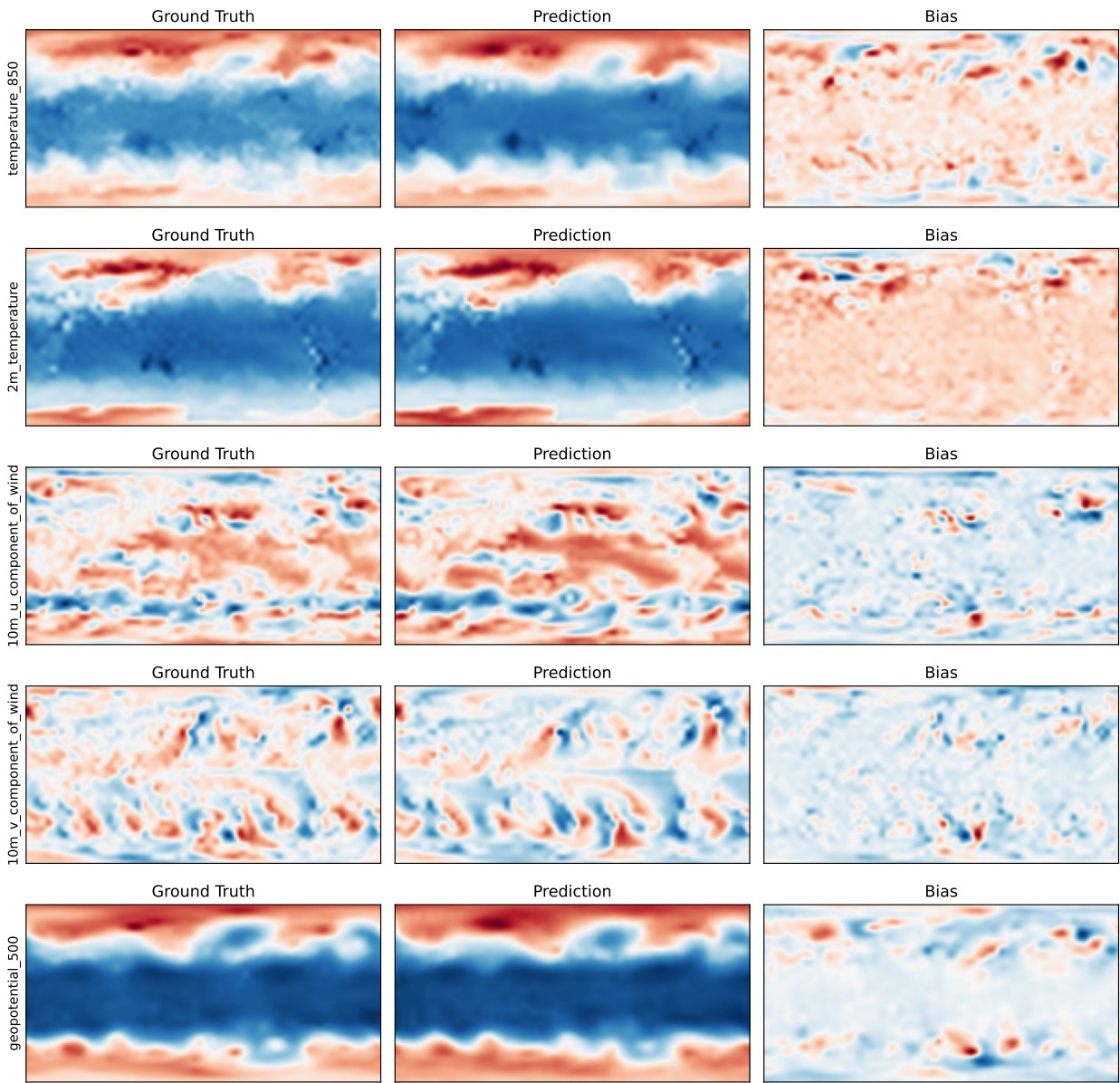

Figure 7: 3 day forecast results of STC-ViT

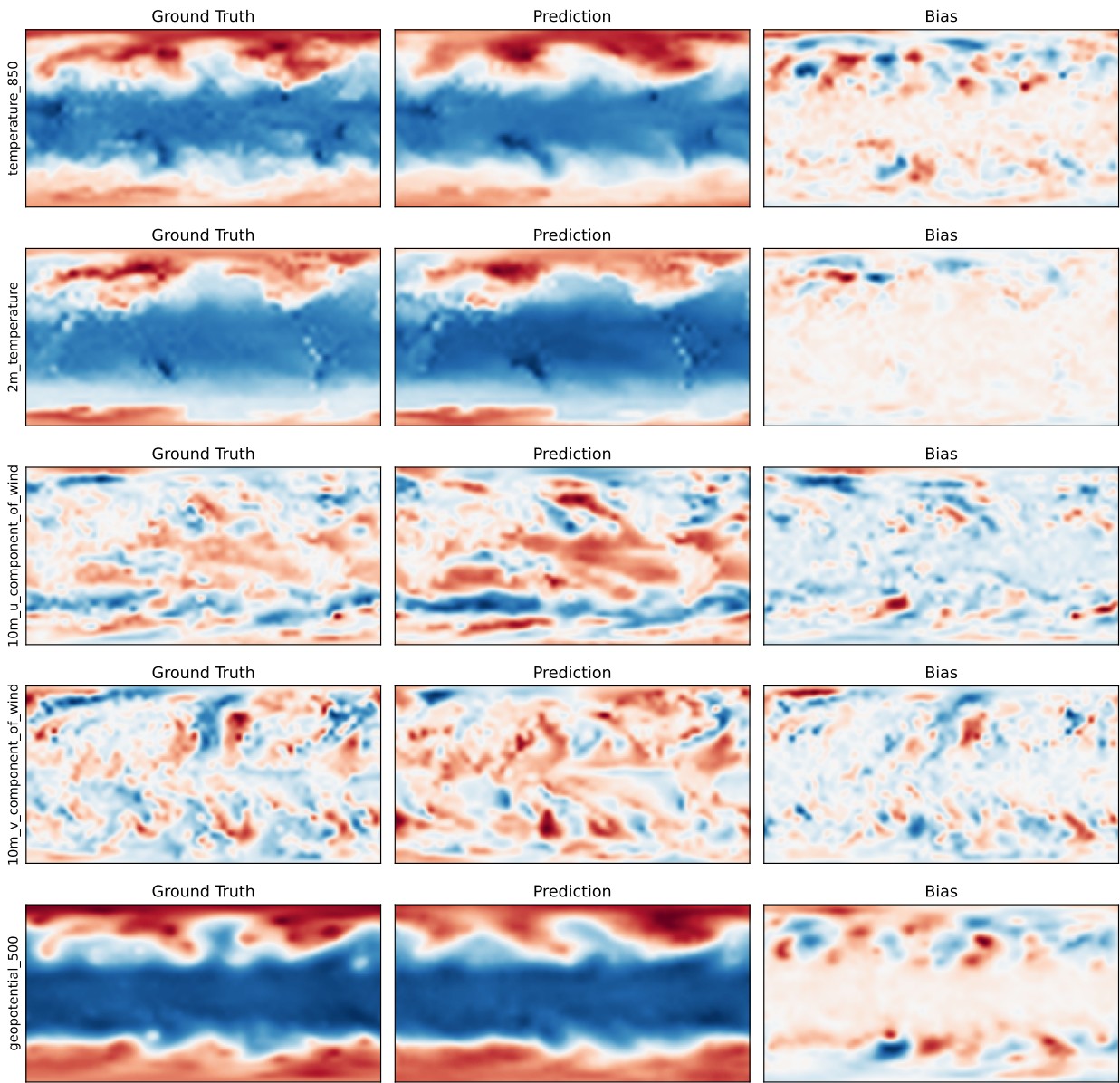

Figure 8: 6 day forecast results of STC-ViT

