# OpenReview forum: "STC-ViT: Spatio Temporal Continuous Vision Transformer for Medium-range Global Weather Forecasting"
_TMLR — Under review for TMLR_

### Review · Reviewer_Txxx · 2026-06-24

**Summary Of Contributions:**

**Summary**

This paper proposes STC-ViT, a global weather forecasting model that integrates **Vision Transformer (ViT)** and **Fourier Neural Operation (FNO)** to model the spatial dependencies, and **Neural Ordinary Differential (NODE)** to continuously model the temporal evolution. The model is evaluated on WeatherBench 1 and 2 under different spatial resolutions, and compared against several existing approaches: ClimODE, ClimaX, Pangu, IFS-HRES, and GraphCast.

**Strengths**
- The paper writing is well-written and easy to follow.
- It demonstrates that STC-ViT with only a single Transformer layer can achieve a competitive performance with several existing models that use deeper Transformer architectures, highlighting its effectiveness. The paper also shows that increasing the Transformer depth can further improve the model performance.
- The introduction of **NODE** serves as a solution for meddling the atmospheric evolution as a continuous-time trajectory, enabling STC-ViT to generate the predictions at different lead times without explicitly training a separate model for each forecast horizon.
- The formulation of the initial condition, $z(0)$ is represented as the combination of data-driven patch embeddings, positional information, Fourier Spectral features and Spherical Harmonic features. This takes Earth-specific geometric information into consideration.

**Weaknesses**
- The comparison with some baselines are not consistent. For example, GraphCast and Pangu-Weather are trained at higher resolution (i.e., $0.25^\circ$), but re-gridded to $1.5^\circ$ resolution for comparison. A consistent comparison with matched resolution would provide a stronger evaluation.
- STC-ViT's performance on the higher resolution datasets (i.e., $1.5^\circ$) appears to be inconsistent across  different lead time. In particular, the model has a worse performance at shorter lead times, which raises the questions that whether the learned continuous dynamics accurately capture fine-scale temporal evolution.
- The paper lacks an ablation study on the components used in Equation 9 to describe the initial latent state, $z(0)$. It would be helpful to evaluate the contribution of every components.

**Audience:**

Yes

**Audience Explanation:**

Some members of TMLR's audience work on this exact topic (global weather forecasting). They may be interested in the paper's findings, particularly the integration of Fourier Spectral Analysis, Spherical Harmonic Analysis and ODE-based solver in weather forecasting systems.

**Broader Impact Concerns:**

No ethical concerns.

**Claims And Evidence:**

Yes

**Claims Explanation:**

The paper claims that the proposed **STC-ViT** is effective and achieves competitive performance relative to existing models. As shown in Table 2, **STC-ViT** requires the least training time and inference time. Furthermore, the results in Table 1 and Figure 2 indicate **STC-ViT** achieves an overall decent performance, although it has a weaker short-term performance than other baselines.

However, the evidence would be stronger if the paper included an ablation study on every components in Equation 9 to describe the initial latent state, $z(0)$, to better understand the contribution of every components. In addition, a more detailed performance on Weatherbench 2 would help clarify **STC-ViT**'s strengths and weaknesses across different forecasting horizons and datasets.

**Requested Changes:**

- Could the authors provide a more detailed performance comparison with all baselines on WeatherBench 1 and 2 (e.g., a full table with all variables and lead times)? This quantitive result would make the comparison easier to analyse.
- Could the authors provide an ablation study on every components in Equation 9 to describe the initial latent state, $z(0)$, to better understand the contribution of every components?
- What dataset and experimental settings are used for the Scaling Analysis?
- Could the authors  clearly the definitions of the sparse and dense time-grid used in the scaling analysis respectively? Since the dense time-grid significantly improves the model's performance, what are the trade-offs that prevent using dense time-grid as the default setting?

---

### Review · Reviewer_g5Gv · 2026-07-17

**Summary Of Contributions:**

**Summary**

This paper proposes a model for weather forecasting that combines FNO layers and a Neural ODE with a ClimaX-style ViT architecture. It aims to overcome the limitations of existing transformer models. Through experiments, the authors claim competitive performance against baselines on the WeatherBench datasets.

**Strengths**

1. The authors propose a model that views the Transformer encoder as a time-conditioned vector field. They also introduce a spherical-harmonic endcoder, which can be a strength in terms of novelty.

2. The paper evaluates on both WeatherBench and WeatherBench 2 following the WB2 convention of scoring on a common 1.5° grid, and the scaling analysis asks the right questions about where the model's capacity comes from.


**Weaknesses**

1. The "what to solve" stated in the authors' introduction is ambiguous. It is difficult to understand whether the premise that transformers are trained separately for each lead time, and the stated problem, actually correspond to the problems of existing related work.

2. The authors' method appears to be nothing more than defining an attention block as the Neural ODE function.

3. The authors provide a code URL, but the actual code is missing from that link.

4. There is an issue that the reproduced ClimODE numbers do not match the published results.

5. In Table 2, V100/TPU/H200 hardware is mixed, and inference times at different resolutions are compared, making the comparison invalid.

6. Figure 1 is inconsistent with the equations in the main text at several points.

7. The symbols L and K collide throughout the paper.

8. The abstract and Broader Impact claim a "low-parameter framework," but the 1.5° model has 156M parameters, larger than GraphCast (37M) and ClimaX (107M).

9. The discussion of prior work is incomplete. Prior studies before ClimODE (2024) that explicitly introduce physical structure or continuous dynamics, as well as the weight-tied / ODE-interpretation transformer line of work, are missing.

10. The first sentence of the introduction cites Couairon et al., 2024 (ArchesWeather), but this citation appears inappropriate, and moreover ArchesWeather is not compared in the experiments.

11. The introduction paragraphs need further improvement.

**Additional Comments:**

The introduction needs careful revision. Beyond the ambiguity of the problem statement (W1 above), several sentences contain inaccurate or unsupported claims (e.g., the per-lead-time training premise, the inappropriate ArchesWeather citation in the first sentence), and the section reference to the results is incorrect (Section 6 points to the Conclusion). I recommend the authors double-check every sentence in the introduction for accuracy and clarity.

**Audience:**

Yes

**Audience Explanation:**

Continuous-time models for data-driven weather forecasting are of active interest to TMLR's audience.

**Claims And Evidence:**

No

**Claims Explanation:**

Not in the current form. The headline claims are not yet supported by the evidence: the reproduced ClimODE numbers do not match the published results (Weakness 4, Requested Change 1), the contribution of the Neural ODE itself is not isolated from simply defining an attention block as the ODE function (Weakness 2, Requested Changes 2 and 6), and the "low-parameter" claim is contradicted by the paper's own 156M parameter count in Table 2 (Weakness 8).

**Requested Changes:**

1. Even with the same 5.625° data and 2017-18 test period as the ClimODE paper, the reproduced numbers differ. Regarding this, you need to report what reproducibility experiments were conducted.

2. Regarding the depth-1 claim, integrating over 6 lead times with Euler instead of RK4 effectively yields a weight-tied depth-6 network. The authors need to further report and discuss performance and effective computation in terms of the number of steps.

4. Regarding the time-conditioned vector field, which part of Equation 11 makes it time-conditioned? The reason I ask this is that it looks like a non-autonomous ODE where time t does not enter the function.

5. Since it is stated as a main contribution in Contribution 2, an ablation study on the PE appears necessary.

6. The authors need to explain how the results compare against pretrained ClimaX as well. In addition, the authors should further explain why comparing against ClimaX retrained with depth 8 constitutes fairness.

7. In Equation 12, the residual of the attention mechanism is used as-is, and during ODE integration there is also a residual structure with the previous state. Was this dual residual architecture intentional?

8. ArchesWeather, cited in the first sentence of the introduction, also needs to be further discussed or compared in the experiments.

9. The sentence in the introduction, "However, transformers in weather forecasting often exhibit structural limitations." does not specify which limitations and should be improved. Beyond this, nearly every sentence in the introduction should be double-checked and written more clearly.

10. The paper presents ClimODE (2024) as essentially the only prior continuous-time approach, but weather and physical-system prediction studies based on physical structure and continuous dynamics have existed before 2024, as listed below (2018~2023). Please discuss these prior works in Section 2.2 (or 3.1) and position STC-ViT within this lineage, based on the design differences from this paper's unconstrained transformer vector field.


> De Bézenac, Emmanuel, Arthur Pajot, and Patrick Gallinari. "Deep learning for physical processes: Incorporating prior scientific knowledge." Journal of Statistical Mechanics: Theory and Experiment 2019.12 (2019): 124009.
>
> Seo, Sungyong, Chuizheng Meng, and Yan Liu. "Physics-aware difference graph networks for sparsely-observed dynamics." International conference on learning representations. 2019.
>
> Iakovlev, Valerii, Markus Heinonen, and Harri Lähdesmäki. "Learning continuous-time pdes from sparse data with graph neural networks." arXiv preprint arXiv:2006.08956 (2020).
>
> Hwang, Jeehyun, et al. "Climate modeling with neural diffusion equations." 2021 IEEE international conference on data mining (ICDM). IEEE, 2021.
>
> Choi, Hwangyong, et al. "Climate modeling with neural advection–diffusion equation." Knowledge and Information Systems 65.6 (2023): 2403-2427.
>
> Dehghani, Mostafa, et al. "Universal transformers." arXiv preprint arXiv:1807.03819 (2018).
>
> Lu, Yiping, et al. "Understanding and improving transformer from a multi-particle dynamic system point of view." arXiv preprint arXiv:1906.02762 (2019).
>
> Li, Bei, et al. "ODE transformer: An ordinary differential equation-inspired model for sequence generation." Proceedings of the 60th Annual Meeting of the Association for Computational Linguistics (Volume 1: Long Papers). 2022.
>
> Bai et al., "Deep Equilibrium Models," NeurIPS 2019

---

### Review · Reviewer_KajL · 2026-07-20

**Summary Of Contributions:**

The Manuscript describes a neural network architecture that instead of operational numerical weather predictor with multiple attention layers to achieves accurate prediction at multiple temporal and spatial scales uses  single  transformer layer to provide to achieve state of the art accuracy. The architecture relies on combination of an neural ODE where a simple first oder differential equation  h'(t) = f(h(t),t,θ)  is numerically solved in continuous time with f being a neural network and  Fourier  Neural Operator for handling different scales that are needed for describing the low and high spatial frequencies of the Kolmogorov energy cascade inherent in Navier - Stokes equations. The architecture contains positional encoding based on spherical harmonics that is a natural embedding for the surface of earth. The time embedding is handled with a set frequencies oscillating in time. The patches of weather data varialble are handled with ViT in a single Transformer Encoders with inputs from the components described above. As recurrent neural networks  (RNN) describe an ODE discretised in time where every time step is one folder layer, Resnet is mathematically an LSTM, where all gates are always open providing the residual connection in time as it is used in the Manuscript - making the time with continuous capability for discretisation with any time step. These provide the capability to create time-conditioned vector fields and that evolve through time with the weather. The result are convincing and support the chosen architecture.

**Additional Comments:**

A very interesting and enlightening  Manuscript.

**Audience:**

Yes

**Audience Explanation:**

The model architecture described is efficient and contains components that addresses the usual difficulties in the neural network based weather predictors. This would certainly interest the audience

The Manuscript addresses a problem that is highly relevant globally.

**Broader Impact Concerns:**

Positive Impact.

**Claims And Evidence:**

Yes

**Claims Explanation:**

The Manuscript is clear in its description of the architectures, it contains ablation analysis, check of the scaling and more importantly provides excellent  results with fast inference time compared to state of the art.

Using approximately equivariant quaternion based Neural ODEs could help in the time evolution of the vector fields as the computation would not face the singularities on the poles compared to the current Spherical Harmonics solution. Also, Icosahedral mappings could be used as patches that are equal across the sphere. This would densify nicely and address the problem in the poles.

**Requested Changes:**

The link https://anonymous.4open.science/r/STCViT-CC8B had only two files in it and I could not see the code. I am interested on the numerical method that is used (non the level, Euler, Runge-Kutta) but on the level of solving a first order ODE for multiple related vector fields that are constrained by conservation laws. Of course, first order derivatives of the a neural network is easy to do, however pure Euler steps have problems with conserved quantities even with analytical values for the derivatives.